# Implications of Cardioprotective Assumptions for National Drinking Guidelines and Alcohol Harm Monitoring Systems

**DOI:** 10.3390/ijerph16244956

**Published:** 2019-12-06

**Authors:** Adam Sherk, William Gilmore, Samuel Churchill, Eveline Lensvelt, Tim Stockwell, Tanya Chikritzhs

**Affiliations:** 1Canadian Institute for Substance Use Research, University of Victoria, Victoria, BC V8P 5C2, Canada; meadowschurchill@gmail.com (S.C.); timstock@uvic.ca (T.S.); 2National Drug Research Institute, Faculty of Health Sciences, Curtin University, Perth, WA 6845, Australia; William.Gilmore@curtin.edu.au (W.G.); Eveline.seeney@gmail.com (E.L.); T.N.Chikritzhs@curtin.edu.au (T.C.)

**Keywords:** alcohol harms, national drinking guidelines, alcohol-attributable deaths, alcohol use and ischaemic heart disease, alcohol policy, International Model of Alcohol Harms and Policies, alcohol’s burden of disease, international comparison, Australia, Canada

## Abstract

The existence and potential level of cardioprotection from alcohol use is contested in alcohol studies. Assumptions regarding the risk relationship between alcohol use and ischaemic heart disease (IHD) are critical when providing advice for national drinking guidelines and for designing alcohol harm monitoring systems. We use three meta-analyses regarding alcohol use and IHD risk to investigate how varying assumptions lead to differential estimates of alcohol-attributable (AA) deaths and weighted relative risk (RR) functions, in Australia and Canada. Alcohol exposure and mortality data were acquired from administrative sources and AA fractions were calculated using the International Model of Alcohol Harms and Policies. We then customized a recent Global Burden of Disease (GBD) analysis to inform drinking guidelines internationally. Australians drink slightly more than Canadians, per person, but are also more likely to identify as lifetime abstainers. Cardioprotective scenarios resulted in substantial differences in estimates of net AA deaths in Australia (between 2933 and 4570) and Canada (between 5179 and 8024), using GBD risk functions for all other alcohol-related conditions. Country-specific weighted RR functions were analyzed to provide advice toward drinking guidelines: Minimum risk was achieved at or below alcohol use levels of 10 g/day ethanol, depending on scenario. Consumption levels resulting in ‘no added’ risk from drinking were found to be between 10 and 15 g/day, by country, gender, and scenario. These recommendations are lower than current guidelines in Australia, Canada, and some other high-income countries: These guidelines may be in need of downward revision.

## 1. Introduction

Alcohol consumption has been proven to cause a significant burden of disease and to be causally linked to dozens of serious health conditions [1,2]. Indeed, drinking was estimated to cause nearly three million deaths globally in 2016; these deaths would not have occurred in the absence of alcohol consumption [1]. In high-income countries, such as Australia and Canada, alcohol-caused death and disability is proportionally greater, as people in these countries drink more alcohol than the global average [1]. National alcohol harm monitoring systems in Australia [3,4] and Canada [5] have revealed a substantial burden of mortality: In Australia there were an estimated 4400 net deaths caused by alcohol in 2015 [4], while in Canada this estimate ranges between 8000 [6] and 14,800 [5].

National and subnational governments may choose to deploy population-wide strategies designed to mitigate this substantial harm. Included in this strategy suite are:(A)Monitoring the extent of the alcohol-caused burden of death and disability through the creation of national alcohol harm monitoring systems [7,8], and(B)publishing national drinking guidelines (NDGs), sometimes called low-risk drinking guidelines, for use by residents to mediate their drinking based on expert-recommended limits [8,9,10].

Australia and Canada each employ both national monitoring systems (the Australian National Alcohol Indicators Project (NAIP) [3] and the Canadian Substance Use Costs and Harms (CSUCH) project [5]) and NDGs. Australian and Canadian NDGs are interesting contrasts. In Australia, NDGs recommend drinking no more than 20 g of pure alcohol per day (g/day), while in Canada the chronic guidelines differ by gender: Men are advised to limit themselves to no more than 28.8 g/day and women no more than 19.2 g/day. We see that daily limits for men are 44% higher in Canada than Australia, while recommendations for women are comparable. In addition, NDGs for both countries are higher than those in some other high-income countries, such as France (14.4 g/day) and the UK (16 g/day). Though the goals of these NDGs are similar, in practice the advice given is quite different; these differences are common globally, as well. For example, among the more than 50 countries that have established NDGs, guidelines related to long-term alcohol use are seen to vary by a factor of nearly six [10]. Further, the 2016 Global Burden of Disease (GBD) for alcohol, using a globally-weighted relative risk (RR) function approach, reported that ‘the level of consumption that minimises health loss is zero’ [2] (p. 1015). In consideration of the marked differences between NDGs at the country level and evidence suggesting a J-shaped curve may not exist, i.e., that alcohol is not health-protective at any drinking level [2,11,12,13], it may be time to reconsider many NDGs worldwide. It has been suggested these weighted RR functions are preferable to all-cause mortality approaches, which have been oft-used in NDG creation, and as function shape is influenced significantly by condition weighting, country-specific analyses should be completed [14]. Australia and Canada share similarities regarding mortality distribution, but also an important difference, as ischaemic heart disease (IHD) is responsible for a higher share of alcohol-related mortality in Australia than in Canada [15,16]; see also Appendix A. We therefore focus on these two countries to provide multiple exemplars across contexts with this important different.

Both national monitoring systems and NDGs are constructed following careful consideration of the assumptions underlying the relationship between alcohol use and the risk of incident morbidity or mortality caused by individual health conditions proven to be caused by drinking. In this regard, for alcohol-caused death in high-income countries, assumptions and methods used to inform the risk relationship between alcohol use and ischaemic heart disease (IHD) are magnified, as IHD is often the leading cause of country-specific mortality [17,18] and hence changes in these decisions can significantly impact findings and related implications. Both monitoring systems and NDGs are often constructed using, as a foundational choice, relative risk (RR) functions relating a measure of average daily alcohol use to the risk of drinker mortality, relative to those who abstain from alcohol or consume little. These RR functions are taken from condition-specific meta-analyses, which combine many studies into single, or gender-specific, risk relationships for each condition. In the case of IHD, three such meta-analyses exist: Those used in the 2016 Global Burden of Disease (GBD) study [2], those used as a basis for the World Health Organization’s Global Status Report on Alcohol and Health (WHO GSRAH) [19], and those used in several national alcohol harms estimations [20], including in Canada [5] and Sweden [21]. The studies differ substantially in function shape, particularly in the amount of cardioprotection (area of the functions below RR = 1.0) offered [2,19,20]. We note that the Zhao et al. study [20] only includes categorical RR relationships in the study; however, an RR function for men was received from the author by special request and was published separately [22].

These three differential, foundational choices, representing the shape of the risk relationship between chronic average alcohol use and IHD mortality delineate three cardioprotective scenarios, which motivate our analyses throughout this study:In Scenario 1, IHD RR functions for men and women were from the 2016 GBD study [2] (p. 97);In Scenario 2, IHD RR functions for men and women were sourced from Roerecke & Rehm [19] (p. 1252), and;In Scenario 3, the IHD RR function for men was sourced from Zhao et al. [20]. The functional form, requested from the author, is a component of a further study [22] (p. 72). The IHD RR function for women was sourced from Roerecke & Rehm [19] (p. 1252); this was chosen as there were too few women-specific studies in the Zhao et al. analysis to complete a gender-specific meta-analysis.

In each scenario, the RR functions for all other alcohol-related health conditions were taken from the 2016 GBD study [2]; this was done to maximize comparability with a well-known estimation of alcohol-attributable mortality. Readers are encouraged to consult these three IHD meta-analyses [2,19,20], as these scenarios are fundamental to the findings of this study.

To our knowledge, a study focusing on the policy implications resulting from these differing cardioprotective scenarios has not been completed. It is likely that these assumptions will substantially affect both estimations of alcohol-caused mortality, i.e., the findings of national alcohol harm monitoring systems, and weighted RR functions, such as those in the GBD study [2], which may then inform NDGs. The aims of this study are therefore to investigate the effect of three cardioprotective scenarios on:Estimates of alcohol-caused mortality in Australia and Canada in 2015, the most recent year for which data was available for both countries, andCountry-specific mortality-weighted RR functions, by gender and total.

We hypothesize that when estimating alcohol-attributable deaths and country- and gender-specific weighted RR functions, the choice of cardioprotective scenario will result in critical differences in two high-income countries and thereby have important implications for NDG revision globally.

## 2. Materials and Methods

### 2.1. Conceptual Framework and Terminology

We use the epidemiological concept of attribution caused by exposure to a harmful substance (in this case alcohol) as the framework for this study [23]. Broadly, the methodology involved estimating the number of alcohol-attributable (AA) deaths in our study populations. Conceptually, this is equivalent to estimating the number of deaths, among conditions for which alcohol has proven causative, which would occur in a parallel society in which all other causative condition factors remain constant, but where population exposure to alcohol is (and always has been) zero. As a thought experiment, we can imagine this as a counterfactual scenario wherein everything is identical, except that alcohol does not exist.

For this project, this is achieved functionally using a health condition-based epidemiological attributable fraction approach [5]; this is the standard method in alcohol epidemiology and is used both globally, e.g., [1,2], and nationally, e.g., [4,5,24]. First, health conditions for which exposure to alcohol has proven to be a contributing factor are identified; we use those conditions identified by the 2016 Global Burden of Disease (GBD) study [2]. Next, alcohol-attributable fractions (AAF) for each condition, by gender and age group, are estimated via the methods presented next. A mortality AAF for liver cancer in men aged 65 and older of 0.25 can be interpreted thus: In a counterfactual scenario where alcohol exposure is zero, 25% of liver cancer deaths in men aged 65+ years would not have occurred in the study year in question. These AAFs are then applied to enumerated mortality in each condition, gender, and age group.

A note on terminology is necessary regarding this study on alcohol and mortality. Throughout the article, we use the terms ‘alcohol-related’ and ‘alcohol-attributable’ often as they are common in the field. They seem similar but are importantly different. As above, alcohol-attributable is specific to the epidemiological framework used, and in this case refers to the number (or fraction) of deaths which would not have occurred in the absence of population exposure to alcohol. In this sense, it is equivalent to stating that alcohol ‘caused’ these deaths, therefore, we use the terms alcohol-attributable and ‘alcohol-caused’ interchangeably. Alcohol-related, however, refers to all deaths caused by a health condition for which alcohol has been proven causative. For example, there were 2758 breast cancer deaths in Australia in 2015 [18]. These deaths are termed alcohol-related, as they accrue to a health condition whose incident and mortality risk is increased by exposure to alcohol [2,25,26]. However, they are not all alcohol-attributable, as an AAF has yet to be applied: The multiplication of this AAF by the condition-specific mortality count will result in the number of alcohol-caused deaths (in each population subgroup). This differentiation is important throughout.

### 2.2. Alcohol-Attributable Fraction Methodology

We use a version of the modern, continuous alcohol-attributable fraction formula [27]. It is modified to follow the methodology used in the most recent GBD study [2], wherein the prevalence of former drinks is ignored and the prevalence of current drinkers in the past year and the prevalence of lifetime abstainers is normalized to sum to 1.0. Conceptually, this is represented as:(1)$$AAF_{c,g,a}=\frac{\int_{0}^{150}P_{g,a}\left(x\right)\left[RR_{c,g}\left(x\right)-1\right]dx}{1+\;\int_{0}^{150}P_{g,a}\left(x\right)\left[RR_{c,g}\left(x\right)-1\right]\;dx}$$where an AAF for each condition, gender, and age group ($AAF_{c,g,a}$) is calculated using a Gamma-based distribution [22,27,28] representing the prevalence of current drinkers ($P_{g,a}\left(x\right)$) and a relative risk function for each condition and gender ($RR_{c,g}\left(x\right)$); each depend on average daily alcohol consumption in grams ethanol per day (x).

Functionally, the calculation is achieved by evaluating the distribution and relative risk functions at integer values between 1 and 150, creating a functional equation of:(2)$$AAF_{c,g,a}=\frac{\sum_{1}^{150}P_{g,a}\left(x\right)\left[RR_{c,g}\left(x\right)-1\right]}{1+\;\sum_{1}^{150}P_{g,a}\left(x\right)\left[RR_{c,g}\left(x\right)-1\right]}$$

We have evaluated the difference between this integer-evaluated formula and integration functions, such as R’s integrate function [29]; the differences are small and in no cases are they larger than 0.5%.

### 2.3. Data Sources

Collected data fulfills the requirements of the above formulations, as well as enumerated mortalities in each health condition, gender, and age grouping. Age groups were 15–34, 35–64 and 65 years and older; these match those used by other alcohol health harms estimations, e.g., [1,2,4,5,30].

#### 2.3.1. Mortality Data

Death data, by health condition, gender, and age group for Australia in 2015 were taken from the Australian Coordinating Registry’s cause of death unit record file dataset [16]. Canadian data for 2015 were taken from Statistics Canada’s CANSIM database [17].

#### 2.3.2. Alcohol Exposure Data

Per capita alcohol sales, for the population 15 years and older, were sourced for Australia from the Australian Bureau of Statistics’ (ABS) estimates based on sales, excise, import, and survey data [31] and for Canada from Statistics Canada’s official sales records [32]. Canadian per capita sales were adjusted upwards using a standard national adjustment to account for unrecorded alcohol, such as that imported or made at home or in make-your-own shops [5,33]. In Australia, the ABS does not adjust for unrecorded consumption, as these sources are considered very low to negligible.

The prevalence of current drinkers, defined as those consuming at least one drink in the past year, and lifetime abstainers for Australia were taken from the National Drug Strategy Household Survey [34] and for Canada were taken from the Canadian Substance Use Exposure Database (CanSUED; [5]), an aggregate database based on Statistics Canada’s 2015 Canadian Tobacco, Alcohol and Drugs Survey (CTADS; [35]).

#### 2.3.3. Relative Risk Functions

Our analyses use the 2016 GBD relative risk (RR) functions [2]. RR functions were obtained from authors of the GBD study and consisted of, for each condition and gender, a vector of values for the relative risk function, and upper and lower 95% confidence intervals, for every 1 g/day increment between 1 and 150 g/day alcohol consumption. Note this matches the necessary input seen in Formula (2).

For ischaemic heart disease (IHD) only, we use two other meta-analyses to inform our choice of RR functions, based on the cardioprotective scenarios detailed in the introduction. These are Roerecke & Rehm [19] and Zhao et al. [20].

#### 2.3.4. Population and Death Data, Totals

We built 2015 population- and death-based rates to compare countries. For Australia, population data were taken from the Australian Bureau of Statistics’ estimated resident population [36] and death data were taken from the Australian Bureau of Statistics [18]. For Canada, population data were sourced from Statistics Canada’s CANSIM database [37] and death data were from Statistics Canada’s Vital Statistics death database [15].

### 2.4. Statistical Analyses

The AAF methodology described above was achieved by employing the International Model of Alcohol Harms and Policies (InterMAHP; [22]), an open access alcohol harms estimator. InterMAHP version 3.0 automates the calculation of Gamma-based prevalence distributions and combines prevalence and relative risk information at integer intervals, for each condition, gender, and age group, as shown in Formula (2). Methods are detailed elsewhere [22]. InterMAHP source code, written in R [29] can be downloaded, then analyzed and modified, if desired.

Further analyses, such as aggregations and the weighting used for weighted RR functions, were completed using R [29]. Figures were completed in Microsoft Excel 2016 (Microsoft Corporation, Redmond, WA, USA).

### 2.5. Estimating Alcohol-Attributable Deaths: Net, Caused and Prevented

In the 2016 GBD study, alcohol was found to have a protective effect, at low or moderate levels of consumption, on three health conditions: Diabetes, ischaemic heart disease, and ischaemic stroke [2]. To estimate deaths caused and prevented by alcohol, as well as net alcohol-attributable deaths, the following steps were followed for Australia and Canada in 2015:Mortality counts, by country, health condition, gender, and age group, were enumerated from official data, see Data Sources. Health conditions were only enumerated if they appeared as the primary cause of death on a mortality record, not if they were listed as a contributing cause; this matches other studies [1,5].Net AAFs, by country, health condition, gender and age group were calculated by InterMAHP using our collected data sources as input, with no modification to the RR functions.Gross AAFs, by country, health condition, gender, and age group were calculated by InterMAHP, but RR functions were modified to have no protection, i.e., all RR values <1.0 were set to 1.0. This has the effect of calculating gross death as any protection is removed.Net and gross alcohol-attributable (AA) deaths were then calculated at this same level of granularity by multiplying both the net and gross AAF of each grouping by the corresponding number of enumerated deaths.Deaths prevented were then calculated as the difference between net deaths and gross deaths.

### 2.6. Creating Weighted Relative Risk Functions, by Gender

Weighted RR functions, first defined in a recent GBD article [2] (p. 1025), weight the RR function for each health condition and gender by the proportion of total alcohol-related mortality that is caused by each health condition. Weighted RR functions were created for Australia and Canada, by gender, using 2015 country-specific mortality weights (see Appendix A for gender-combined weights). For example, IHD is the alcohol-related condition which causes the highest mortality burden in both Australia (36.2% of alcohol-related mortality) and Canada (32.4% of alcohol-related mortality). Country- and gender-specific functions were weighted by corresponding country-gender weightings.

## 3. Results

### 3.1. Per Capita Consumption and Drinking Prevalence in Australia and Canada

Per capita alcohol consumption and the prevalence of current and former drinkers and lifetime abstainers are shown in Table 1. Per person drinking, among those 15 years and over, is slightly higher in Australia (9.90 L per year) than in Canada (9.69 L per year). Differences were seen in the prevalence of the population who identify as lifetime abstainers (higher in all six population subgroups in Australia than in Canada) and as former drinkers (higher in Canada than Australia).

### 3.2. Net, Gross, and Prevented Alcohol-Attributable Deaths under Three Cardioprotective Scenarios

Table 2 presents AA deaths, by condition category, health condition, and country, under three cardioprotective scenarios. In 2015, we estimate that alcohol was causally responsible for between 2933 and 4570 deaths in Australia and between 5179 and 8024 deaths in Canada, depending on the meta-analytic source of the RR function used for IHD.

In Australia, under each of the three scenarios, cancer was the leading categorical cause of death; there were an estimated 1349 AA cancer deaths. Scenarios had a major impact on AA deaths accruing to IHD, as the Global Burden of Disease methodology resulted in a net figure of 1303 deaths prevented, Roerecke & Rehm resulted in 335 net deaths prevented and Zhao with Roerecke & Rehm resulted in 333 net deaths caused. In the last scenario, this results in cardiovascular conditions causing the second-most AA deaths (1063), by category, while in the first scenario the cardiovascular category contributes 574 net deaths prevented, which is least harmful (most protective) among the seven health condition categories. Aside from IHD, diabetes (126 deaths prevented) and ischaemic stroke (IS; 49 deaths prevented) were health conditions for which alcohol provided net protection from death.

By health condition in Australia, alcohol-caused cirrhosis of the liver was the leading cause of net AA deaths (850). In each of Scenario 2 and 3, IHD was the leading cause of gross AA deaths, but those caused-deaths were offset by estimated deaths prevented from moderate consumption.

In Canada, the leading categorical cause of net AA deaths was also cancer (2332). However, regardless of cardioprotective scenario, the second- and third-ranked categories in Canada were digestive conditions (1950 deaths caused) and injuries (1782 deaths caused). Depending on scenario, cardiovascular conditions were then ranked between fourth (Scenario 3; 1127 deaths caused) and last (Scenario 1; 1718 deaths prevented). By health condition, cirrhosis of the liver was the leading cause of net AA deaths (1843) and the leading cause of gross AA deaths in all but Scenario 2 (IHD: 1925 deaths caused). In Scenario 2, these gross deaths caused were more than offset by a large number of deaths prevented (3231), resulting in 1306 net deaths prevented by alcohol due to IHD.

### 3.3. Population- and Death-Based Rates of Net Alcohol-Attributable Deaths

Population- and death-based rates of net alcohol-caused deaths in Australia and Canada are presented in Table 3, in order to facilitate country comparisons. Incident deaths rates, per 1,000,000 population, are shown in the top panel. We found Canada had higher rates of net AA death for five of seven condition categories, save for cardiovascular conditions and diabetes. In each of three cardioprotective scenarios, Australia was estimated to have higher rates of net alcohol-caused cardiovascular death.

Results per 10,000 deaths, as opposed to population denominators, were largely similar. Australia was estimated to have higher death-based rates in the same two condition categories, while Canada had experienced higher rates in the other five condition categories.

### 3.4. Net Alcohol-Attributable Deaths, by Gender and Health Condition Category

Net AA deaths, by gender and condition category, are presented in Table 4. In both countries and in each scenario, men experience the vast majority of net AA deaths: Ranging between a low of 76.3% of gender-combined mortality in Australia under Scenario 2 to a high of 95.3% in Canada under Scenario 1. In Australia, among men in Scenarios 1 and 2, the leading categorical causes of net AA deaths were cancer, injuries, and digestive conditions. In Scenario 3, cardiovascular conditions caused the most net AA deaths. Canadian men experience AA mortality from similar condition categories, as digestive conditions, injuries and cancer were the leading causes of death in Scenarios 1 and 2. Similarly to Australia, in Scenario 3 cardiovascular conditions were instead the leading cause of net AA deaths, based on a large number of alcohol-caused IHD mortalities estimated in that scenario.

Women were not exempt from harm; cancer was the leading cause of net AA deaths for women in both Australia (541 deaths caused) and Canada (935 deaths caused). In both countries, digestive conditions and injuries caused a significant burden of AA mortality. Under all scenarios, Canadian women experienced a higher magnitude of cardioprotection than Australian women, even considering Canada’s larger population.

### 3.5. Weighted Relative Risk Functions, by Cardioprotective Scenario

Figure 1 shows RR functions, weighted by gender-combined mortality in Australia (Figure 1A,C) and Canada (Figure 1B,D). Gender-combined mortality weights are shown in Appendix A. Weighted RR functions, for each of three cardioprotective scenarios, are similar in Australia and Canada, with small differences in the range of 0 to 20 g/day. Using GBD risk functions in Scenario 1, the weighted RR function achieves a minimum at 10 g/day in both Australia (RR = 0.976) and Canada (RR = 0.986). These exposure levels represent the estimated theoretical minimum risk exposure level (TMREL). In Australia, this RR function remains below RR = 1.0 until 13 g/day; while in Canada it is 15 g/day. At higher levels of average daily drinking, all three functions behave similarly in the two countries, with Scenario 2 functions reaching the highest level of risk at 150 g/day (RR nearly 6.0), followed by Scenario 3 (RR ~ 5.5) and Scenario 1 (RR just less than 4.0).

In both countries, Scenario 2 provided the largest area under RR = 1.0, i.e., the most protection from low levels of average drinking. In Australia, the Scenario 2 function reaches a minimum at 5 g/day and remains protective until 14 g/day, while in Canada this function achieves a minimum at 7 g/day and is protective until 16 g/day. Scenario 3 functions were shown to have little area below RR = 1.0, in Australia and Canada the RR minimum is at 3 and 5 g/day and the functions are protective until 6 and 10 g/day, respectively.

### 3.6. Weighted Relative Risk Functions, by Gender

Figure 2 depicts RR functions, weighted by gender-specific mortality, using GBD cardioprotective assumptions (Scenario 1). In both Australia and Canada, at all levels of average daily drinking, men were found to experience a higher RR than women; however the TMREL for both genders is seen to be 10 g/day (Figure 2C,D). In Australia, the weighted RR function for women remains below RR = 1.0 until 15 g/day, while for men this value is 12 g/day. In Canada, the function is protective until 18 g/day for women and 13 g/day for men.

Appendix A depict gender-specific weighted RR functions, by the three cardioprotective scenarios. In Australia, women reach minimum risk at 6 g/day and ’no added’ risk at 14 g/day in Scenarios 2 and 3, while men under Scenario 2 reach a nadir at 5 g/day and experience protection until 14 g/day. In Canada, women reach the TMREL at 7 g/day and ‘no added’ risk at 17 g/day in Scenarios 2 and 3. Men achieve minimum risk at 7 g/day and protection until 15 g/day in Scenario 2. In Scenario 3, men in both countries experience no protection at any level of consumption, i.e., the weighted RR functions is monotonically increasing from 0 g/day.

### 3.7. Weighted Relative Risk Functions, by Net vs. Gross Harms

Weighted RR functions, representing Australian gross vs. net harm functions, are shown in Appendix A for each cardioprotective scenario. Gross harm functions represent only the lives lost to alcohol use and none of the prevented deaths, i.e., condition-specific functions are set to RR = 1.0 when the function is protective. There are no conditions for which alcohol is protective past approximately 60 g/day in all scenarios, and indeed we see the gross and net RR functions equalize around this point (61 g/day in Scenario 1; 63 g/day in Scenario 2 and 54 g/day in Scenario 3). By definition, the gross RR function cannot be protective and presents a TMREL of 0 g/day. The gross and net RR functions show the most difference in Scenarios 1 and 2 and little difference in Scenario 3. Gender-specific functions depicting net vs. gross weighted RR functions are shown in Appendix A.

## 4. Discussion

Our hypothesis, that choice of cardioprotective scenario will produce critical differences in Australia and Canada when estimating alcohol-attributable deaths and country- and gender-specific weighted RR functions, was supported. These findings have important implications for national alcohol harm monitoring systems and national drinking guidelines (NDGs). Across three cardioprotective scenarios, estimates of AA deaths differed by as much as 57% in Australia and 55% in Canada: This difference was due solely to assumptions underlying the risk relationship between drinking and IHD. However, in both countries and under all scenarios, drinking was responsible for a substantial burden of preventable mortality. Alcohol use was estimated to cause between 2933 and 4570 deaths in Australia and between 5179 and 8024 net deaths in Canada.

In comparing alcohol-caused mortality in the two countries using population- and death-based rates from Table 3, it is seen that Canada experiences slightly higher population-based mortality rates than Australia under each scenario and higher death-based rates in two of three scenarios. This is somewhat surprising, given similar current drinking patterns and the fact that Australian per capita drinking is slightly higher than that observed in Canada. However, on the other hand, nearly twice as many (13%) Canadians identify as former drinkers than Australians (7%). As former drinkers experience significant accumulated alcohol harms [1], it is likely that this explains this result.

The ranges of AA deaths estimated in our three cardioprotective scenarios compare differentially to existing national monitoring systems in Australia and Canada. In Australia, the National Alcohol Indicators Project estimated 4410 net AA deaths in 2015 [4], within the range of estimates produced by our study scenarios. However, in Canada, the Canadian Substance Use Costs and Harms (CSUCH) project estimated over 14,800 alcohol-caused deaths in 2014, far above the estimates produced by the GBD RR functions used here, under each of the three scenarios. The 2018 WHO GSRAH [1], which calculates the 2016 alcohol-caused burden, reported 7272 AA deaths in Australia and 11,509 AA deaths in Canada [38]. We note that these estimates, produced using different RR functions for all alcohol-related health conditions in this study (save for IHD in some scenarios), are considerably higher than the estimates presented herein and, for Canada, more in line with those reported by CSUCH. Clearly, choices made regarding which RR functions to apply for each health condition are enormously influential on burden of disease estimates and must be a key consideration when designing alcohol harm monitoring systems.

The mortality-weighted relative risk (RR) functions presented here, under three cardioprotective scenarios and in two high-income countries, provide insight into potentially necessary revisions for NDGs. Such functions may be used to discern the average daily alcohol consumption level that would:(A)Minimize risk, i.e., the theoretical minimum risk exposure level (TMREL);(B)equalize the risk of a drinker with that of a non-drinker, i.e., result in ‘no added risk’ from alcohol use, or;(C)be used to inform higher, ‘low risk drinking guidelines’, which advise guidelines based on a chosen level of ‘acceptable additional risk’ from alcohol use.

The last can be completed using various methodologies, e.g., through calculating alcohol exposure levels resulting in a defined level of increased lifetime mortality risk (such as 1 in 100 or 1 in 1000), as has been done in previous studies [39]. Indeed, 2009 Australian guidelines were designed using a 1% increase in lifetime mortality [39,40]. As (C) is subjective by definition and determination of acceptable levels of risk for populations requires consideration be given to a broad range of social and cultural factors specific to that population; we encourage analysts and decision-makers engaged in drinking guidelines and recommendations to complete this exercise independent of our efforts. We therefore focus our discussion and recommendations using (A) and (B) above, as these are scientifically defined.

A synthesis of our figures leads us to provide the following advice towards (A) and (B). If a gender-combined recommendation is preferred and depending on cardioprotective scenario, minimum risk (TMREL) is achieved at between 3 and 10 g/day in Australia and between 5 and 10 g/day in Canada. If guideline goals were instead to choose a consumption level resulting in no added risk, a drinking level of between 6 and 14 g/day in Australia and between 10 and 16 g/day in Canada would be forwarded. If gender-differentiated guidelines were forwarded, under Scenario 1, men and women each achieve minimum risk at 10 g/day in both countries, while ‘no added’ risk occurs for men at 12 or 13 g/day and women at 15 or 18 g/day in Australia and Canada, respectively. We note that in light of the substantial differences caused by cardioprotective scenario for both minimum and no added risk levels, it is challenging to provide recommendations given the current uncertainty regarding the risk relationship between alcohol use and IHD. Typically, in regard to human health, the most conservative choice would be taken from among multiple options, so as not to provide potentially dangerous advice. In this respect, the low levels of consumption resulting in minimum (3–5 g/day) and no added (10 g/day) risk would be found from Scenario 3, which assumes the least cardioprotection from alcohol use. We note that the analyses presented here use only mortality weights; to fully advise NDGs, these analyses may be replicated using a composite measure of morbidity and mortality, such as Disability Adjusted Life Years (DALYs).

A further important result is seen in comparing gender-specific weighted RR functions in Figure 2. In both countries and for all average consumption levels, men experience a higher relative risk due to drinking. This result is at odds with previous research largely showing that women experience greater risk at a given drinking level than men [19,41,42]. However, our results are unambiguous: Global Burden of Disease risk functions and mortality distributions in Australia and Canada show men are exposed to higher risk for any level of daily drinking. We note that only four GBD risk functions are gender-differential: Those for IHD, ischaemic stroke (IS), haemorrhagic stroke (HS), and diabetes. Gender differences are therefore necessarily driven by this subset of functions, which provide greater protection to women, in the case of IHD, IS and HS; this provides explanation as to why women experience lower risk at low consumption levels. At high consumption levels, an explanation is forwarded in the form of breast cancer, which causes a high burden of mortality among women in both countries (about 10%) and whose risk function has a low RR for all levels of consumption.

Towards NDGs overall, it is important to note that compared to any of the above recommendations, both Australian and Canadian NDGs, as currently formulated, are high (20 g/day in Australia and 19.2 g/day for women and 28.8 g/day for men in Canada). Policymakers may wish to review and revise national guidelines to better advise populations regarding their alcohol consumption; this extends to high-income countries globally [9]. At a minimum, NDGs should provide information regarding the TMREL, so that those who choose to drink may endeavour to minimize their risk of death accruing to an alcohol-related condition due to their alcohol use.

## 5. Limitations

This study suffers from limitations. Both alcohol harm estimation and weighted RR function specification are dependent on RR functions from meta-analyses that are not specific in context or time. Meta-analyses are invariably composed of studies from across many cultures, countries (largely high-income nations) and time periods, i.e., RR functions are used which may not be representative of the country and time period in question. We apply these functions to Australia and Canada as a best available estimate, as country-specific functions do not exist, and we note that all alcohol harms estimations suffer from this limitation.

GBD 2016 methodology separates former drinkers into lifetime abstainers and current drinkers, by normalizing the prevalence of the latter two to sum to 1.0 [2]. This is unusual in alcohol epidemiology, as most national [3,5] and global estimations [1,30,43] include explicit harms estimation for those who used to drink. This methodology is new and its effect on alcohol harms estimation should be studied in detail, preferably before another GBD analysis is undertaken. Next, estimation of drinking occasion-based alcohol-caused injuries from distributional methods using average daily drinking may present a source of uncertainty. Nevertheless, in the absence of a viable alternative this method is used widely [1,2,5,21].

The weighted RR function that appeared in the 2016 GBD of alcohol study [2] used global Disability Adjusted Life Years (DALYs) to weight conditions, while our analyses use mortality weighting. This may have the effect of increasing the weight of some causes of death, such as IHD, which are more likely to occur later in life. As IHD assumptions were the particular focus of our analyses, this may exacerbate the differences between our mortality-weighted RR functions, as compared to analogous analyses using DALYs. Next, DALYs may provide a better composite of AA harms than mortality alone. Regarding advice for NDGs, we note that our results are not disaggregated by racial and ethnic differences; these differences may lead to disparities in alcohol-caused mortality [44]—this is an area for future work.

Last, as this study was a collaborative effort of teams in Australia and Canada, using a set of risk functions from the GBD, we could only include alcohol-related health conditions used by all teams. This will lead to underestimation of total alcohol-caused mortality, as some conditions used by each team were not included. However, the number of deaths excluded is small.

## 6. Conclusions

As IHD is the leading cause of mortality in many high-income countries, differential assumptions regarding the risk relationship between alcohol use and IHD are hugely influential in estimating alcohol-caused mortality and weighted RR functions. AA mortality in three scenarios using different IHD risk estimates differed by more than 50% for both Australia and Canada. National alcohol harm monitoring systems may benefit from sensitivity analyses accounting for these differences. However, in all scenarios and both countries, alcohol was causally responsible for a significant burden of mortality.

To inform NDGs, country-specific weighted RR functions are recommended, as the global distribution of disease burden is dissimilar to most country profiles; this is especially true of high-income countries. Our analyses show that, in relation to alcohol use, the minimum risk of mortality from health conditions related to drinking occurs at or below an average daily consumption level of 10 g/day pure alcohol. If NDGs instead advise an alignment of risk between drinkers and abstainers, our study suggests daily levels of between 10 and 15 g/day in most scenarios and both genders. Both results are lower than current NDGs in Australia, Canada, and many other countries worldwide.

## Figures and Tables

**Figure 1 ijerph-16-04956-f001:**
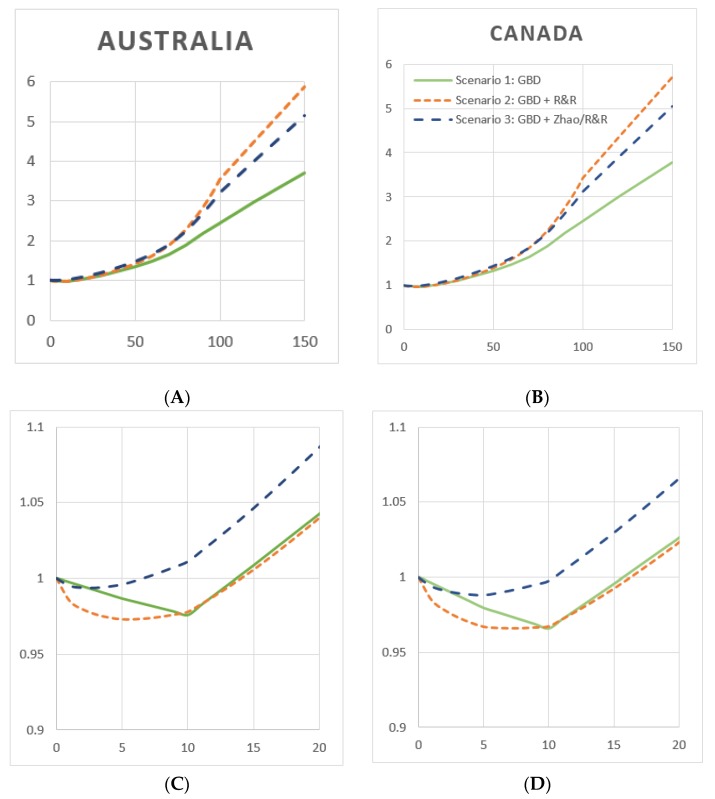
Weighted relative risk functions, by cardioprotective scenario, Australia and Canada, 2015. (**A**) Australia, net harm, genders combined. (**B**) Canada, net harm, genders combined. (**C**) Same functions as in (**A**), magnified to a range of 0–20 g/day for resolution. (**D**) Same functions as in (**B**), magnified to a range of 0–20 g/day for resolution. **Y-axis**: Relative risk; **x-axis**: Average daily alcohol consumption in grams ethanol/day. (**Green line**) Scenario 1: GBD; (**orange line**) Scenario 2: GBD + R&R; (**blue line**) Scenario 3: GBD + Zhao/R&R.

**Figure 2 ijerph-16-04956-f002:**
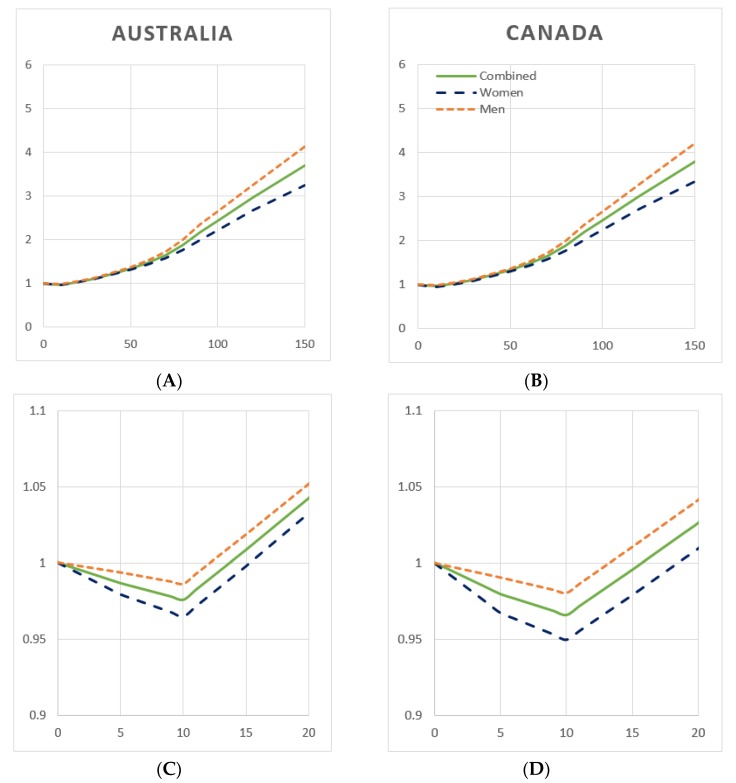
Weighted relative risk functions, by gender, Australia and Canada, 2015. (**A**) Australia, net harm, cardioprotective scenario 1: GBD. (**B**) Canada, net harm, cardioprotective Scenario 1: GBD. (**C**) Same functions as in (**A**), magnified to a range of 0–20 g/day for resolution. (**D**) Same functions as in (**B**), magnified to a range of 0–20 g/day for resolution. **Y-axis**: Relative risk; **x-axis**: Average daily alcohol consumption in grams ethanol/day. (**Green line**) combined; (**blue line**) women; (**orange line**) men.

**Table 1 ijerph-16-04956-t001:** Per capita alcohol consumption and prevalence of current drinkers, former drinkers and lifetime abstainers, Australia and Canada.

Gender	Age Group	Australia (APC = 9.90 L)			Canada (APC = 9.69 L)		
		Current Drinkers	Former Drinkers	Lifetime Abstainers	Current Drinkers	Former Drinkers	Lifetime Abstainers
Male	15 to 34	0.75	0.05	0.21	0.81	0.07	0.12
	35 to 64	0.85	0.06	0.08	0.82	0.12	0.06
	65+	0.80	0.13	0.08	0.74	0.20	0.05
Female	15 to 34	0.72	0.05	0.23	0.75	0.09	0.17
	35 to 64	0.81	0.08	0.11	0.77	0.14	0.09
	65+	0.68	0.14	0.18	0.65	0.24	0.12
Total		0.78	0.07	0.15	0.77	0.13	0.10

APC = alcohol per capita (population 15+ years old); L = litres ethanol; rows may not sum to 1.0 due to rounding.

**Table 2 ijerph-16-04956-t002:** Alcohol-attributable net, gross and prevented deaths under three cardioprotective scenarios, Australia and Canada, 2015.

Health Category and Condition	Australia			Canada		
	Scen. 1: GBD	Scen. 2: GBD + R&R	Scen. 3: GBD + Zhao/R&R	Scen. 1: GBD	Scen. 2: GBD + R&R	Scen. 3: GBD + Zhao/R&R
**Cancers**						
Breast cancer	293.1			528.9		
Colorectal cancer	430.8			854.8		
Laryngeal cancer	51.6			94.7		
Liver cancer	161.4			262.8		
Oesophageal cancer	136.2			207.2		
Oropharyngeal cancer	275.6			383.5		
Subtotal	1348.6			2331.8		
**Cardiovascular Conditions**						
Atrial fibrillation and flutter	171.8			313.8		
Haemorrhagic stroke	280.8			418.4		
Hypertensive heart disease	271.7			417.8		
Ischaemic heart disease						
Deaths caused:	186.0	1268.6	900.7	295.7	1925.4	1414.5
Deaths prevented:	1489.0	1603.7	567.3	2877.4	3231.4	1151.3
Net deaths:	−1303.0	−335.1	333.4	−2581.6	−1306.0	262.3
Ischaemic stroke	Deaths caused: 54.9			Deaths caused: 262.5		
	Deaths prevented: 103.9			Deaths prevented: 628.4		
	Net deaths: −49.0			Net deaths: −365.9		
Alcoholic cardiomyopathy	54.0			80.0		
Subtotal						
Deaths caused:	1019.4	2102.0	1734.1	1788.5	3418.2	2907.3
Deaths prevented:	1593.2	1707.8	671.4	3506.1	3860.2	1780.1
Net deaths:	−573.8	394.2	1062.7	−1717.6	−442.0	1127.2
**Diabetes**						
Subtotal	Deaths caused: 35.6			Deaths caused: 110.8		
	Deaths prevented: 162.0			Deaths prevented: 569.6		
	Net deaths: −126.4			Net deaths: −458.8		
**Digestive Conditions**						
Cirrhosis of the liver	849.6			1842.6		
Pancreatitis	45.8			106.9		
Subtotal	895.4			1949.6		
**Infectious Diseases**						
Tuberculosis	14.3			33.1		
Lower respiratory infections	145.3			315.1		
Subtotal	159.6			348.3		
**Neuropsychiatric Conditions**						
Alcohol use disorders	248.0			888.0		
Epilepsy	54.2			56.3		
Subtotal	302.2			944.3		
**Injuries**						
Alcohol poisoning	68.0			281.0		
Interpersonal violence	27.4			63.4		
Self-harm	501			780.0		
Transport injuries	147.9			276.1		
Unintentional injuries	183.2			381.3		
Subtotal	927.5			1781.9		
Grand Total						
Deaths caused:	4688.3	5770.9	5403.0	9255.0	10,884.7	10,373,8
Deaths prevented:	1755.2	1869.0	833.5	4075.8	4429.8	2349.7
Net deaths:	2933.1	3901.0	4569.5	5179.2	6454.9	8024.1

Scen. = Scenario; GBD = Global Burden of Disease; R&R = Roerecke & Rehm. The number of alcohol-attributable deaths are the same for all health conditions, except ischaemic heart disease, in each of the three scenarios.

**Table 3 ijerph-16-04956-t003:** Population- and death-based rates of net alcohol-attributable death, Australia and Canada, 2015.

Health Category	Australia			Canada		
	Scen 1: GBD	Scen 2: GBD + R&R	Scen 3: GBD + Zhao/R&R	Scen 1: GBD	Scen 2: GBD + R&R	Scen 3: GBD + Zhao/R&R
**Rate per 1,000,000 Population**						
Cancer	71.9			78.0		
Cardiovascular conditions	−30.6	21.0	56.6	−57.4	−14.8	37.7
Ischaemic heart disease	−69.4	−17.9	17.8	−86.3	−43.7	8.8
Diabetes	−6.7			−15.3		
Digestive conditions	47.7			65.2		
Infectious diseases	8.5			11.6		
Neuropsychiatric conditions	16.1			31.6		
Injuries	49.4			59.6		
Total	156.3	207.9	243.5	173.2	215.8	268.3
**Rate per 10,000 Deaths**						
Cancer	86.6			89.0		
Cardiovascular conditions	−36.8	25.3	68.2	−65.6	−16.9	43.0
Ischaemic heart disease	−83.6	−21.5	21.4	−98.6	−49.9	10.0
Diabetes	−8.1			−17.5		
Digestive conditions	57.5			74.4		
Infectious diseases	10.2			13.3		
Neuropsychiatric conditions	19.4			36.1		
Injuries	59.5			68.0		
Total	188.3	250.4	293.3	197.7	246.4	306.4

Scen. = Scenario; GBD = Global Burden of Disease; R&R = Roerecke & Rehm. Ischaemic heart disease (IHD) rates are a subset of cardiovascular conditions; IHD is therefore right justified.

**Table 4 ijerph-16-04956-t004:** Net alcohol-attributable deaths, by gender and health condition category, Australia and Canada, 2015.

Health Category	Australia			Canada		
	Scen 1: GBD	Scen 2: GBD + R&R	Scen 3: GBD + Zhao/R&R	Scen 1: GBD	Scen 2: GBD + R&R	Scen 3: GBD + Zhao/R&R
**Men**						
Cancer	807.3			1,397.1		
Cardiovascular conditions	−140.4	371.5	1040.0	−404.4	324.0	1893.2
Ischaemic heart disease	−667.8	−156	512.6	−1362.3	−633.9	935.2
Diabetes	2.5			−5.3		
Digestive conditions	705.5			1549.9		
Infectious diseases	119.3			271.9		
Neuropsychiatric conditions	237.5			727.4		
Injuries	733.9			1399.9		
Subtotal	2465.5	2977.4	3645.9	4936.4	5664.8	7234.0
Percent of total	84.1%	76.3%	79.8%	95.3%	87.8%	90.2%
**Women**						
Cancer	541.4			934.7		
Cardiovascular conditions	−433.4	22.7	22.7	−1313.3	−766.0	−766.0
Ischaemic heart disease	−635.2	−179.1	−179.1	−1219.4	−672.1	−672.1
Diabetes	−128.9			−453.5		
Digestive conditions	189.9			399.6		
Infectious diseases	40.3			76.4		
Neuropsychiatric conditions	64.7			216.9		
Injuries	193.6			382.0		
Subtotal	467.6	923.7	923.7	242.8	790.1	790.1
Percent of total	15.9%	23.7%	20.2%	4.7%	12.2%	9.8%
**Both Genders**						
Cancer	1348.6			2331.8		
Cardiovascular conditions	−573.8	394.2	1062.7	−1717.6	−442.0	1127.2
Ischaemic heart disease	−1303.0	−335.1	333.4	−2581.6	−1306.0	263.2
Diabetes	−126.4			−458.8		
Digestive conditions	895.4			1949.4		
Infectious diseases	159.6			348.3		
Neuropsychiatric conditions	302.2			944.3		
Injuries	927.5			1781.9		
Grand Total	2933.1	3901.0	4569.5	5179.2	6454.9	8024.1

Scen. = Scenario; GBD = Global Burden of Disease; R&R = Roerecke & Rehm. Ischaemic heart disease (IHD) rates are a subset of cardiovascular conditions; IHD is right justified. ‘Percent of total’ values are calculated column-wise.

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
