# Peer review of "Implications of Cardioprotective Assumptions for National Drinking Guidelines and Alcohol Harm Monitoring Systems"

_ijerph, 2019, doi:10.3390/ijerph16244956_

Round 1

Reviewer 1 Report

This is an interesting and valuable study of alcohol-related epidemiology. Using three previous meta-analysis studies, the authors investigated how varying assumptions lead to differential estimates of alcohol-attributable deaths and weighed relative risk functions. They concluded that consumption levels resulting in no added risk from drinking are lower than current guidelines in Australia and Canada. This conclusion is important for prevention of alcohol-induced disease and reduction of alcohol-related deaths. There are some issues that are recommended to be clarified before recommendation of this paper to publication as follows:

The reason why Australians and Canadians were compared in relation to alcohol-induced deaths in this study is unclear because they are racially similar. Please add possible explanations of the reasons for the differences in the results of Australians and Canadians to Discussion. In addition, it is preferable to add speculated racial and ethnic differences in low risk drinking guidelines. In Figures 1 and 2, there were very small decreases in relative risk of light drinkers compared with nondrinkers. Are they significant or not? Do the authors think very low alcohol consumption has no preventive effect on total deaths? Regarding the weighed relative risk function for women and men, women and men remained relative risk of 1.0 until 15 and 12 g/day, respectively, in Australia and until 18 and 13 g/day, respectively, in Canada (Figure 2). These findings may mean that women are more resistant to alcohol than men and disagree with the general concept of gender difference in alcohol resistance. How do the authors explain these gender difference in this study? Table 3: How are the upper and lower panels different? Maybe men and women? Please clarify this. Figures 1 and 2: Please add legends to x and y axes. Please adjust the format of tables to that of the journal.

Author Response

Reviewer #1

(R)eviewer’s comment: This is an interesting and valuable study of alcohol-related epidemiology. Using three previous meta-analysis studies, the authors investigated how varying assumptions lead to differential estimates of alcohol-attributable deaths and weighed relative risk functions. They concluded that consumption levels resulting in no added risk from drinking are lower than current guidelines in Australia and Canada. This conclusion is important for prevention of alcohol-induced disease and reduction of alcohol-related deaths. There are some issues that are recommended to be clarified before recommendation of this paper to publication as follows:

(A)uthor’s response: Thank you for the comment and for your willingness to review this manuscript.

(R)eviewer’s comment: The reason why Australians and Canadians were compared in relation to alcohol-induced deaths in this study is unclear because they are racially similar. Please add possible explanations of the reasons for the differences in the results of Australians and Canadians to Discussion.

(A)uthor’s response: A good point. We add two sentences to the second paragraph in the Introduction to motivate this. We note that ischaemic heart disease (our focus here) is responsible for a substantially higher share of Australian mortality than Canadian; hence motivating the use of these two countries.

Next, we have added a second paragraph to the Discussion regarding the observed country-specific differences in population- and death-based mortality rates.

R: In addition, it is preferable to add speculated racial and ethnic differences in low risk drinking guidelines.

A: We agree. We have added treatment of this issue in the Limitations section, as we were unable to study this in our current study.

R: In Figures 1 and 2, there were very small decreases in relative risk of light drinkers compared with nondrinkers. Are they significant or not?

A: As we received the RR functions from the IHME Global Burden of Disease group (and they do not have confidence intervals), we cannot state whether the functions below 1.0 in a  statistically significant manner at any consumption level g/day.

R: Do the authors think very low alcohol consumption has no preventive effect on total deaths?

A: The weighted RR function analyses show a theoretical minimum risk exposure level (TMREL) of between 3 g/day and 10 g/day across countries and scenarios. Our analyses in this article suggest that alcohol-caused mortalities would be minimized if the entire drinking population drank exactly this amount each day. Of course, this may not be practical, but this is the conclusion of the analysis, as we state.

Our Discussion regarding the TMREL in the article is, we believe, an answer to your question: total deaths would be slightly lower in a scenario where everyone drank the TMREL. However, as above, since we received the GBD functions from IHME, we cannot tell if this is statistically significant.

R: Regarding the weighted relative risk function for women and men, women and men remained relative risk of 1.0 until 15 and 12 g/day, respectively, in Australia and until 18 and 13 g/day, respectively, in Canada (Figure 2). These findings may mean that women are more resistant to alcohol than men and disagree with the general concept of gender difference in alcohol resistance. How do the authors explain these gender difference in this study?

A: An excellent point. In fact, for all levels of consumption, the weighted GBD risk functions show that men experience greater harm than women. You are correct that this is disparate with many previous findings. We have added a full new paragraph in the discussion (second to last), explaining possible causes.

R: Table 3: How are the upper and lower panels different? Maybe men and women? Please clarify this.

A: The top panel is ‘Rate per 1,000,000 population’ and the bottom is ‘Rate per 10,000’ deaths, ie. The table presents population- and death-based rates to compare the two countries. These are labeled above the health categories.

R: Figures 1 and 2: Please add legends to x and y axes.

A: We had originally labeled the axes; however, since both are four panels, it was repetitious to label all four. Also, doing so shrunk the figure considerably, since all four would have labels. This is why we decided to drop the legend down into the notes, so the figures would be more readable in the publication.

R: Please adjust the format of tables to that of the journal.

A: We have spent considerable time formatting the tables to fit the journal style, to the best of our ability. The table in the journal template is small, and so does not provide much information toward creating larger tables.

We are willing to work with the print edit team to bring them closer to the journal style, though we are not what sure to do at this stage.

Reviewer 2 Report

I have no major concerns. The study demonstrates (for one disease , IHD) that AA mortality is quite sensitive to changes in underlying RR functions. I believe that is not really surprising. There are many studies pointing at caveats of using the attributable fraction approach. Especially the interplay between the relative risks of different aa diseases might not be addressed properly, i.e. it is possible that when accounting for the other relative risks say for cancers as well the total sums of aa deaths might converge under the different scenarios; this would be the result of differently accruing comorbidities in the background literature (i.e. WHO GBD 2016, RR, Zhao et al.). 

However I think the study is an eye-opener especially for public health organizations that need to present information on alcohol attributable mortality (and morbidity) risks to health politics. For example if a specific political program focusses on IHDs alone or a medical association is only interested in specific disease subgroups according to their specialisation.

Concerning national drinking guidelines, it should be added that recommendations are an aggregate of a lot of gathered evidence. For example in DALY calculations, years living with disabilities would matter, economic costs matter, (co-)morbidities matter. Please make this clear in your discussion: a recommendation can only be given if an aggregate measure on aa morbidity and mortality can be calculated not from looking at one disease alone.

Author Response

Reviewer #2

(R)eviewer’s comments: I have no major concerns. The study demonstrates (for one disease , IHD) that AA mortality is quite sensitive to changes in underlying RR functions. I believe that is not really surprising. There are many studies pointing at caveats of using the attributable fraction approach. Especially the interplay between the relative risks of different aa diseases might not be addressed properly, i.e. it is possible that when accounting for the other relative risks say for cancers as well the total sums of aa deaths might converge under the different scenarios; this would be the result of differently accruing comorbidities in the background literature (i.e. WHO GBD 2016, RR, Zhao et al.). 

However I think the study is an eye-opener especially for public health organizations that need to present information on alcohol attributable mortality (and morbidity) risks to health politics. For example if a specific political program focusses on IHDs alone or a medical association is only interested in specific disease subgroups according to their specialisation.

(A)uthor’s response: Thank you for your comments and for your review.

R: Concerning national drinking guidelines, it should be added that recommendations are an aggregate of a lot of gathered evidence. For example in DALY calculations, years living with disabilities would matter, economic costs matter, (co-)morbidities matter. Please make this clear in your discussion: a recommendation can only be given if an aggregate measure on aa morbidity and mortality can be calculated not from looking at one disease alone.

A: A good point. We have added treatment of this at the end of the fifth paragraph in the Discussion (sentence begin with ‘We note that the analyses….’).

Also, we have added a sentence in the limitations section, third paragraph, begins with ‘Next, DALYs…..’.